# The Impact of Intra-City and Inter-City Innovation Networks on City Economic Growth: A Case Study of the Yangtze River Delta in China

Xianzhong Cao [ID], Bo Chen, Yi Guo and Zhenzhen Yi *

The Center for Modern Chinese City Studies, East China Normal University, Shanghai 200062, China; xzcao@geo.ecnu.edu.cn (X.C.); 51203902028@stu.ecnu.edu.cn (B.C.); 52203902001@stu.ecnu.edu.cn (Y.G.)
* Correspondence: zzyi@iud.ecnu.edu.cn

**Abstract:** Innovation networks promote regional innovation and economic growth. Using the patent data of cooperative inventions and the panel data of socio-economic statistics for 2010–2019, this study quantitatively analyzes the spatial structure evolution of intra-city and inter-city innovation networks for 41 cities in the Yangtze River Delta and their influence on economic growth. This study shows that these networks are increasingly connected and have a highly similar Z-shaped spatial structure. City economic growth is generally high, relatively stable, and mainly positively influenced by inter-city innovation networks. Intra-city innovation networks have no significant effect on economic growth; however, they are complementary to the inter-city ones.

**Keywords:** innovation networks; economic growth; spatio-temporal evolution; influence mechanism; Yangtze river delta

## 1. Introduction

Innovation network refers to the cooperation of innovation actors, such as government, enterprises, universities, research institutions, and intermediary service agencies in technology research and development [1]. Considering cities as boundaries, innovation networks can be divided into intra-city and inter-city ones. With the economic globalization, the innovation paradigm has changed from the traditional closed linear model to the modern open network one, and the influence of innovation networks on regional innovation and economic growth has gradually become the frontier scientific problem for economic geographers [2,3]. As the representative of developed regions in developing countries, The Yangtze River Delta has experienced rapid economic development and strong innovation potential, and the Chinese central government attaches great importance to the construction of a regional innovation community. On 20 December 2020, the Ministry of Science and Technology in China released the Development Plan for the Construction of Innovation Community in the Yangtze River Delta, which clearly pointed out that it was necessary to focus on high-tech industries, such as biomedicine, new materials, integrated circuits, equipment manufacturing, and to achieve cross-border cooperation with innovation actors, such as universities, research institutions, enterprises, and intermediaries in 41 cites, so as to promote the free flow of innovation elements. All these make it a typical region for studying intra-city and inter-city innovation networks; however, little research has focused on the relationship between innovation networks and economic growth, especially in terms of the former influencing the latter.

Economic geographers have studied the structure characteristics, spatial scale, influence mechanism, and effect of innovation networks. Among them, the research on innovation networks in different spatial scales, such as global, local, and global–local, has attracted the attention of many scholars; however, such research generally focuses on the analysis of the network structure rather than on the relationship mechanisms, and the

influence of innovation networks of different spatial scales (intra-city and inter-city) on economic growth remains controversial [4]. Therefore, taking 41 cities in the Yangtze River Delta as examples, this paper analyzes the spatial and temporal evolution characteristics of intra-city and inter-city innovation networks and economic growth, trying to answer the following three questions: first, what are the characteristics of intra-city and inter-city innovation networks in the Yangtze River Delta and the differences between them? Second, what are the characteristics of the spatial and temporal evolution of economic growth in the Yangtze River Delta? Third, what is the impact of intra-city and inter-city innovation networks on economic growth in the Yangtze River Delta and the impact mechanism?

The remainder of this paper is organized as follows: the subsequent section presents the literature review. Section 3 describes the data and empirical variables used in analyzing the innovation networks and economic growth. With the help of ArcGIS 10.6 and MaxDEA, Section 4 analyzes the spatio-temporal evolution of innovation networks and economic growth in the Yangtze River Delta. Based on the regression model, Section 5 discusses the mechanism of innovation networks' impact on economic growth. The final section concludes and discusses the paper.

## 2. Literature Review

### 2.1. Relationship between Innovation Networks and Economic Growth

New knowledge based on technological innovation is recognized by scholars as the foundation of promoting economic growth [5,6]. On the basis of the traditional economic growth theory, the endogenous economic growth theory emphasizes the key role of knowledge in driving productivity and economic growth [7]. It also points out that economic growth and innovation networks are interrelated, as knowledge creation, accumulation, and transfer can effectively explain the differences in city economic growth levels [8]. The innovation network is an important way to promote regional growth [9]. The purpose of innovation networks is to improve the innovation ability and performance of innovation actors, and then to transform the research and development of innovative products into economic benefits. Innovation can be regarded as one of the necessary conditions for regional economic growth and development. The existing research has also shown that the knowledge flow in innovation networks determines technological innovation ability and the level of regional economic growth [10].

Innovation networks strengthen the knowledge flow inside and outside the region and is a key capital investment in the process of regional economic growth [11]. However, there is still controversy about the relationship between innovation networks of different spatial scales and economic growth. Local knowledge creation and global knowledge acquisition interact with regional economic growth [12], and we can see knowledge integrators with high competitiveness innovate by integrating global and local knowledge [13]. Crespo mainly analyzed the influence of the structural attributes of local knowledge networks on the promotion of regional competitiveness [14], while Breschi et al. pointed out that non-local innovation networks are more conducive to regional economic development [15]. There are considerable differences in the influence of different spatial scales of knowledge on regional development; the course of regional economy and innovation not only depends on localized production and knowledge creation, but also needs to combine the "local buzz" and "global pipelines" [16–18]. At the same time, it should be emphasized that there are costs for innovation actors to cooperate with local and non-local innovators. Esposito et al. found that too much local interaction would lead to the disappearance of the boundaries of innovation actors and the decline in regional technological innovation, when innovation actors engage in non-local interaction, the costs may also exceed the benefits [12]. Bianchi et al. observed that acting as interregional broker cities, especially connecting Latin American cities to the rest of the world, negatively affected patent outcomes [19].

*2.2. Influence of Innovation Networks on City Economic Growth*

City economic growth is the ultimate embodiment of the economic effect of innovation networks, and innovation is the necessary condition to maintain stable economic growth and development. Existing research results show that knowledge flow in innovation networks determines innovation ability and economic growth [10]. The position and rights of actors in innovation networks are the important factors that affect its knowledge acquisition and control the knowledge flow. Bianchi et al. found that cities in the center of innovation networks had more innovative activities, while as regional "gatekeepers", accessing and using external knowledge required costs and internal capabilities [19]. Moreover, cities acting as regional brokers have a negative influence on their own innovation capability due to the effects of "information overload" and "mobilization failure" [20]. On the contrary, Le Gallo et al. used collaborative patent data in the biomedical industry to find that gatekeepers who directly acquired non-local knowledge could improve innovation performance and benefit the entire region [21]. The influence of proximity among innovation agents in innovation networks has also attracted scholars' attention, including geographical, cognitive, social, institutional, and cultural proximity [22]; proximity significantly promotes the formation of innovation networks in an ITISA and contributes to the improvement of innovation ability [23]. In addition, the selection of knowledge sources in innovation networks is also important for regional economic growth. Balland et al. found that what matters is not being connected to other regions per se, but being connected to regions that provide complementary capabilities [24]. Regional knowledge absorption capacity moderates the impact of innovation networks on regional economic growth. The lack of regional knowledge absorption and utilization capacity leads to the evolution of local knowledge into path lock [25], which eventually reduces the power of regional economic growth. Some scholars pointed out that the characteristics of knowledge determine the network value, which in turn affects the economic growth [11]. The structural attribute of innovation networks modulates the effect of the knowledge flow, which determines the innovation capability and performance of regions or enterprises. Regional development is restricted by the innovation capability of the respective regions or enterprises, and areas occupying the core position in innovation networks are more likely to bring about regional economic growth [26].

Generally speaking, compared to the role of traditional economic factors in city economic growth, knowledge flow and innovation have become the key driving factors for city economic growth and productivity improvement [27]. However, the related research needs to study in more depth the relationship mechanism; the existing research pays more attention to the relationship between technological innovation and economic growth, or regards the innovation network as a whole but pays little attention to the relationship between intra-regional and inter-regional innovation networks and economic growth. Therefore, this study analyzed the differences in the spatial and temporal evolution of economic growth from the perspective of innovation networks and compared the influence of intra-city and inter-city networks, exploring their influence mechanisms.

## 3. Methods and Data

*3.1. Research Sample*

The Yangtze River Delta is one of the most dynamic, open, and innovative regions in China. In 2020, the GDP of the Yangtze River Delta was CNY 3851 billion, accounting for 24.1 percent of the whole country's GDP. High-quality and regional integration have become the core solutions for the Yangtze River Delta. On 30 December 2020, the Ministry of Science and Technology in China issued the Development Plan for the Construction of Science and Technology Innovation Community in the Yangtze River Delta, further clarifying the advantages of regional innovation resources, optimizing the regional innovation layout and collaborative innovation ecology, enhancing the regional collaborative innovation capability, and supporting the high-quality development and regional integration of the Yangtze River Delta. According to the Regional Integration Development Plan of the

Yangtze River Delta issued by the government in December 2019, the research sample of this paper included the whole area of three provinces and one municipality in the Yangtze River Delta (Shanghai, Jiangsu, Zhejiang, and Anhui provinces, covering 258,000 square kilometers), with 41 cities.

*3.2. Calculation of Innovation Networks in the Yangtze River Delta*

Innovation can be regarded as one of the necessary conditions for regional economic growth and development. However, with the increasing complexity and uncertainty of knowledge innovation, an actor's innovation can no longer be satisfied only by its own resources. The purpose of the innovation network is to improve the innovation ability and performance of innovation actors, and then to transform the research and development of innovative products into economic benefits. Essentially, innovation networks are the connections between different innovation actors, including government, enterprises, universities, research institutions, and intermediary service agencies, while the urban innovation network classifies the cooperation links to the urban level.

In this study, we constructed multi-level innovation networks, including intra-city and inter-city innovation networks. Collaboration among the innovators within a city formed the intra-city innovation networks. Each node represents an innovator and the edges represent their collaborative relations. Similarly, collaborations among the cities formed the inter-city innovation networks. Each node represents a city and the edges represent collaborative relations among cities. Considering that different innovation actors within a city belong to the same city, we used the number of collaborations between innovation actors to measure the intensity of collaborative innovations within the city. Moreover, we used the social network analysis (SNA) to analyze innovation networks. Degree centrality is one of the indicators to measure the centrality of the nodes in the network, and its calculation formula is as follows:

$$DC = \sum_{j=1}^{n} X_{ij} \tag{1}$$

where n is the number of cities in the network, $X_{ij}$ is the level of cooperation between *i* and *j*. The higher the centrality, the better the position of a city in sharing, integrating, and utilizing complementary and heterogeneous resources.

The patent literature is the largest source of technical information in the world. The report by the World Intellectual Property Organization (WIPO) shows that about 90–95 percent of global R&D outputs is contained in patents, and the rest is embodied in scientific literature, such as papers and publications [28]. The patent literature offers the advantages of openness, timeliness, detailed content, and easy comparisons between industrial technologies or different spaces, and has become an important data source for studying knowledge production and innovation activities [29]. A joint patent application is an interactive innovation process based on knowledge sharing between organizations and resource integration embedded in social networks. Economic geographers have widely recognized that joint patent application data can be researched on innovation networks and knowledge spillovers [30]; the invention patent represents the original technology, which can better reflect the technological innovation achievements. The patent data are sourced from the incoPat database and processed as follows: (i) time selection—because it takes 18 months for patent applications to be published, our research limited the patent application dates to the period from 1 January 2010 to 31 December 2019, and the locations to Shanghai, Anhui, Zhejiang, and Jiangsu province. The patents with 1 number of applicants were filtered, and individuals were filtered in the applicant category to obtain 256,934 joint patent applications in the Yangtze River Delta. (ii) Matching the innovation actors and geographical location—we matched the geographical positions of the applying actors by using the enterprise database of Tianyancha (https://www.tianyancha.com/search, accessed on 20 June 2021) and cross-processed the data of three or more actors, thus obtaining 400,201 linkages of city innovation networks. (iii) The classification of innovation networks—we divided the innovation networks within the Yangtze River Delta into intra-city and inter-city

ones, and finally obtained 161,766 linkages of intra-city and 86,487 linkages of inter-city innovation networks (Figure 1).

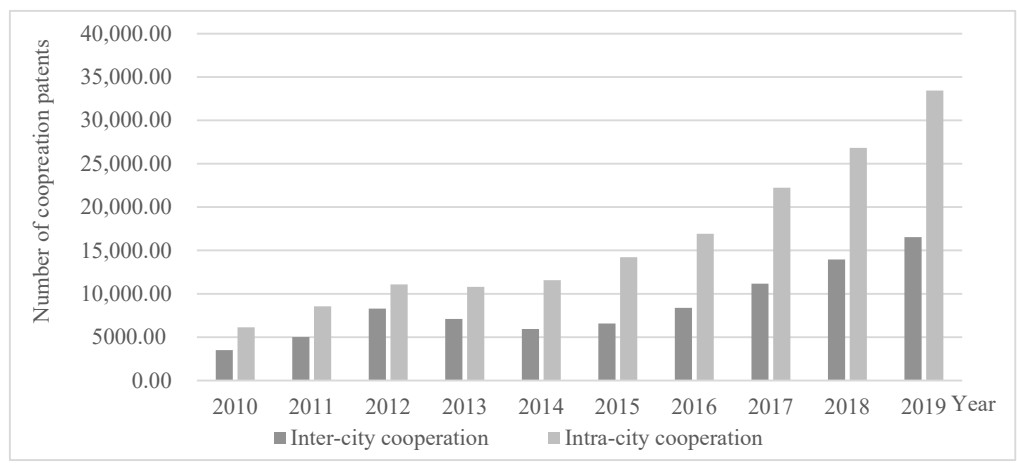

**Figure 1.** Diagram of the intra-city and inter-city innovation cooperation networks in the Yangtze River Delta from 2010 to 2019.

### 3.3. Calculation Method of Economic Growth in the Yangtze River Delta

3.3.1. Total Factor Productivity of DEA Malmquist Index

The total factor productivity (TFP), which is generally accepted by scholars, is used as an index to measure city economic growth, while the TFP of the DEA Malmquist index is mostly used as the measurement method [31]. Based on the relative efficiency, Data Envelopment Analysis (DEA) evaluates its relative effectiveness by comparing the degree of deviation of (decision-making unit: DMU) from the DEA frontier, which is a non-parametric method. The advantage of this method is that it does not need any specific function form or distribution assumption. It is effective in dealing with the efficiency problems of multi-input and multi-output DMUs and is suitable for the efficiency evaluation of urban complex economic systems. Therefore, based on the method of the DEA Malmquist index, this paper analyzed the dynamic characteristics of economic growth in the Yangtze River Delta.

The Malmquist productivity index is based on the DEA model, which uses the ratio of the distance function to calculate the input–output efficiency. The Malmquist productivity index given to output index variables is:

$$M_0^t = \frac{D_0^t(x_{t+1}, y_{t+1})}{D_0^t(x_t, y_t)}; \ M_0^{t+1} = \frac{D_0^{t+1}(x_{t+1}, y_{t+1})}{D_0^{t+1}(x_t, y_t)} \tag{2}$$

$D_0^t(x_t, y_t)$ and $D_0^{t+1}(x_{t+1}, y_{t+1})$ are the output distance functions over the same periods of t and t + 1; $D_0^t(x_{t+1}, y_{t+1})$ and $D_0^{t+1}(x_t, y_t)$ are the output distance functions over the mixed periods of t and t + 1.

Fare et al. calculated the Malmquist index of the fixed 4e direction output with the geometric average of the two Malmquist productivity indices [32], which can be written in the following equivalent form:

$$M_0(x_t, y_t, x_{t+1}, y_{t+1}) = \left[ \frac{D_0^t(x_{t+1}, y_{t+1})}{D_0^t(x_t, y_t)} * \frac{D_0^{t+1}(x_{t+1}, y_{t+1})}{D_0^{t+1}(x_t, y_t)} \right]^{1/2} \tag{3}$$

Further decomposition:

$$M_0(x_t, y_t, x_{t+1}, y_{t+1}) = \frac{D_0^{t+1}(x_{t+1}, y_{t+1})}{D_0^t(x_t, y_t)} * \left[ \frac{D_0^t(x_{t+1}, y_{t+1})}{D_0^{t+1}(x_{t+1}, y_{t+1})} * \frac{D_0^t(x_t, y_t)}{D_0^{t+1}(x_t, y_t)} \right]^{1/2} \tag{4}$$

Formula (3) is a variant of Formula (2), which was used to express the separation of technical change and technical efficiency change, that is, MI = EF * TC. Furthermore, the change in technical efficiency can be further divided into pure technical efficiency change (PEC) and scale efficiency change (SEC), that is, MI = PEC * SEC * TC.

### 3.3.2. The Input–Output Index System

This paper studied the TFP of cities as a decision-making unit, which is based on the (Cobb–Douglas: C–D) production function model in economic growth theory. The essence of national and regional economic development is the input of capital, land, labor, and other production factors. According to the neoclassical economic growth theory and the understanding of economic growth in cities and urban agglomerations [33], this study selected capital, land, and labor as the input indicators and urban economic aggregates as the output indicators. The data sources and processing methods of each variable were described in continuation(Table 1):

**Table 1.** Selection and explanation of the growth input and output indexes in the Yangtze River Delta.

| Indicator Type | Secondary Indicator | Measure Indicators |
|---|---|---|
| Input indicator | Capital | Social fixed-assets investment |
|  | Land | Built-up area of each city |
|  | Labor | The sum of employees in urban units and private and individual employees |
| Output indicator | The economic output | GDP |

First, input variables:

(i) capital variables. Scholars often use the perpetual inventory method of social fixed-assets investment to estimate the capital stock. The formula is: $K_{it} = K_{it-1}(1 - \delta) + I_{it}/p_t$, where $\delta$ is the depreciation rate; $p_t$ is the fixed assets the investment price index. For the capital stock in the base year, this study used the material capital stock of three provinces and one municipality in the Yangtze River Delta in 2000, and calculated the material capital stock of cities in the Yangtze River Delta in the following way. The depreciation rate $\delta$ was 9.6 percent, and the fixed-assets investment price index referred to China's fixed-assets investment price published annually by the National Bureau of Statistics (https://data.stats.gov.cn/easyquery.htm?cn=C01, accessed on 21 June 2021); the data of new fixed-assets investments in each city was obtained from the China Urban Statistical Yearbook.

$$\frac{\text{The material capital stock of cities in 2000}}{\text{The material capital stock of provinces in 2000}} = \frac{\text{GDP of cities in 2000}}{\text{GDP of provinces in 2000}} \quad (5)$$

(ii) Labor variables. The number of employees in each city was obtained and the specific indicators were characterized by the sum of employees in urban units and private and individual employees.

(iii) Land variables. We selected the built-up area of each city to represent the input of urban land elements.

Second, output variables: the total annual GDP of each city was selected to represent the economic output of the city.

## 4. Results Analysis

### 4.1. Characteristics of Innovation Networks in the Yangtze River Delta Region

Based on the joint patent applications of the cities in the Yangtze River Delta from 2010 to 2019, ArcGIS 10.6 software was used to describe the spatiotemporal evolution of innovation networks within and between these cities(Figure 2).

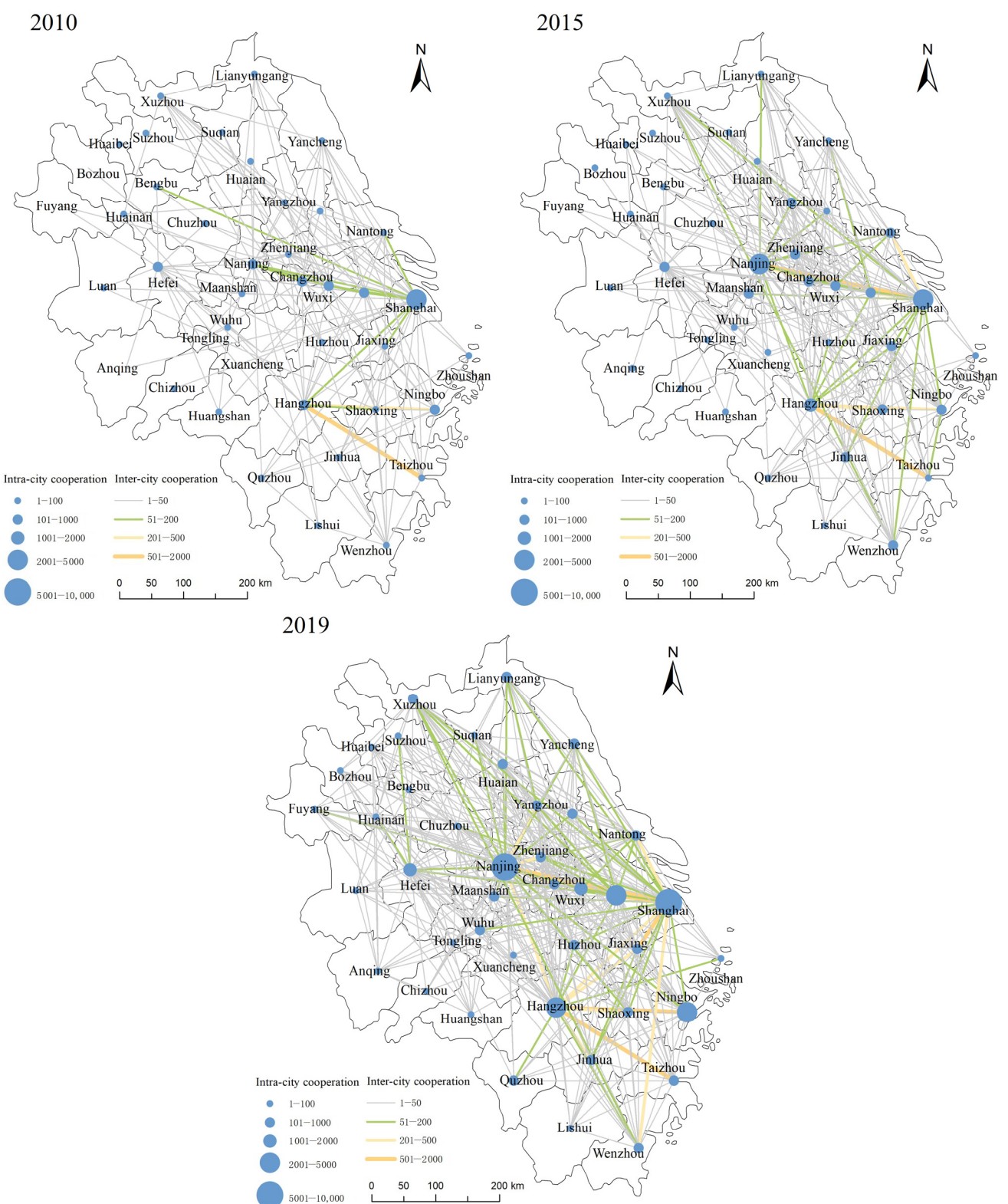

**Figure 2.** Illustration of the intra-city and inter-city innovation networks in the Yangtze River Delta from 2010 to 2019.

The intra-city innovation networks of the Yangtze River Delta present a "core—periphery" structure, and the core changed from a single center in Shanghai to multi-centers "Nanjing–Shanghai–Hangzhou–Ningbo". This result also accords with the studies of Rombach et al. [34] and Zhang et al. [35], and refers to their measurements of network

structure. Specifically, the highlands of the intra-city innovation cooperation are mainly concentrated in the core cities of Shanghai, Jiangsu, and Zhejiang. Anhui, southwest of Zhejiang and north of Jiangsu, has become a depression for intra-city cooperation. In 2010, Shanghai became the single core of the Yangtze River Delta with 2736 intra-city innovation collaborations. Hangzhou was far behind, with only 902 intra-city innovation collaborations; the difference between the two was more than three-times greater. In 2019, Shanghai and Nanjing became the cores with 7191 and 6488 intra-city innovation collaborations, respectively. In addition, Hangzhou, Ningbo, and Suzhou each had 4007, 3176, and 3088 intra-city innovation collaborations as the sub-core. Only Hefei in Anhui entered the top ten with 1686 intra-city collaborations. From the perspective of growth and changes, from 2010 to 2019, intra-city innovation cooperation in 41 cities in the Yangtze River Delta was on the rise, with the core cities maintaining a growth rate of 3 to 10 times; Nanjing experienced an increase of nearly 10 times in the past 10 years, totaling 5915 intra-city innovative collaborations, while the peripheral cities, such as Bozhou, Fuyang, Suqian, or Lishui, due to the relatively small base of innovation cooperations within the city in 2010, experienced a significant increase in 10 years. However, there is still a big gap between the absolute amount of growth and the core cities.

The inter-city innovation networks of the Yangtze River Delta also present a similar "core–periphery" structure, while being more compact. Specifically, cities with strong innovation capabilities, such as Hangzhou, Shanghai, and Nanjing, have become the core of the inter-city innovation networks, and other cities in Anhui province, except Hefei, with less inter-city cooperation, have become the edge of the network. This spatial distribution is highly similar to the spatial pattern of inter-city technology flow in the Yangtze River Delta based on the patent transfer network observed in the existing studies. The inter-city innovation cooperation pairs also increased from 152 in 2010 to 361 in 2019. The scale of the main body of innovation cooperation between cities is also increasing accordingly; from the number of "partner cities", it can be observed that the average scale of the top-ten city pairs in 2010 was 227, which increased nearly 3 times, to 667, in 2019 (Table 2). The centrality of the inter-city innovation networks also increased from 3.55% in 2010 to 10.03% in 2019, an increase of nearly 3 times, which fully shows that these networks are becoming more closely connected; at the same time, the core is also improving. The status of core cities in inter-city innovation networks is becoming more and more important.

**Table 2.** Comparison of the top-ten innovation cooperation scales within and between cities in the Yangtze River Delta in 2010 and 2019.

| 2010 | | 2019 | |
|---|---|---|---|
| **Partner Cities** | **Cooperation Scale (Pieces)** | **Partner Cities** | **Cooperation Scale (Pieces)** |
| Hangzhou–Taizhou | 1149 | Hangzhou–Ningbo | 1327 |
| Hangzhou–Ningbo | 213 | Shanghai–Suzhou | 1118 |
| Hangzhou–Shanghai | 179 | Hangzhou–Taizhou | 785 |
| Shanghai–Suzhou | 178 | Jiaxing–Shanghai | 610 |
| Jiaxing–Shanghai | 125 | Nanjing–Suzhou | 594 |
| Nanjing–Shanghai | 117 | Nanjing–Shanghai | 562 |
| Shanghai–Wuxi | 98 | Hangzhou–Jinhua | 443 |
| Suzhou–Wuxi | 79 | Nanjing–Yangzhou | 421 |
| Nantong–Shanghai | 77 | Hangzhou–Shaoxing | 414 |
| Hangzhou–Shaoxing | 59 | Hangzhou–Shanghai | 394 |

Overall, the spatial structure of intra-city and inter-city innovation networks in the Yangtze River Delta is relatively consistent; over time, the "Z-shaped" core–periphery structure is gradually appearing. Shanghai, Nanjing, and Hangzhou became the core nodes of intra-city and inter-city innovation networks in the region; Suzhou, Ningbo, Hefei, and Wuxi became the sub-core nodes, which is consistent with the spatial structure of

the "local—cross-border" innovation network in the Yangtze River Delta cities based on cooperative patent data.

### 4.2. Temporal and Spatial Evolution Characteristics of Economic Growth in the Yangtze River Delta

Based on the endogenous growth model, urban capital stock, labor force, and land were selected as the input indicators, and urban economic aggregates as the output indicators. We used MaxDEA software to calculate the TFP of the 41 cities from 2010 to 2019 and to describe the spatiotemporal evolution of urban economic growth.

Figure 3 shows the changes in the average TFP of cities in Shanghai, Jiangsu, Zhejiang, and Anhui from 2010 to 2019. We found that Shanghai had the highest total factor productivity, followed by Jiangsu and Zhejiang, and Anhui was the lowest; Shanghai, Jiangsu, and Zhejiang were higher than the regional average. Anhui became a depression for economic development in the Yangtze River Delta region. From the perspective of dynamic trends, the overall TFP of the Yangtze River Delta had relatively stable changes; while the TFP of Shanghai fluctuated greatly, showing an overall upward trend, Anhui and Zhejiang generally showed a trend of rising first and then decreasing. Jiangsu's TFP showed a slight upward trend in fluctuations.

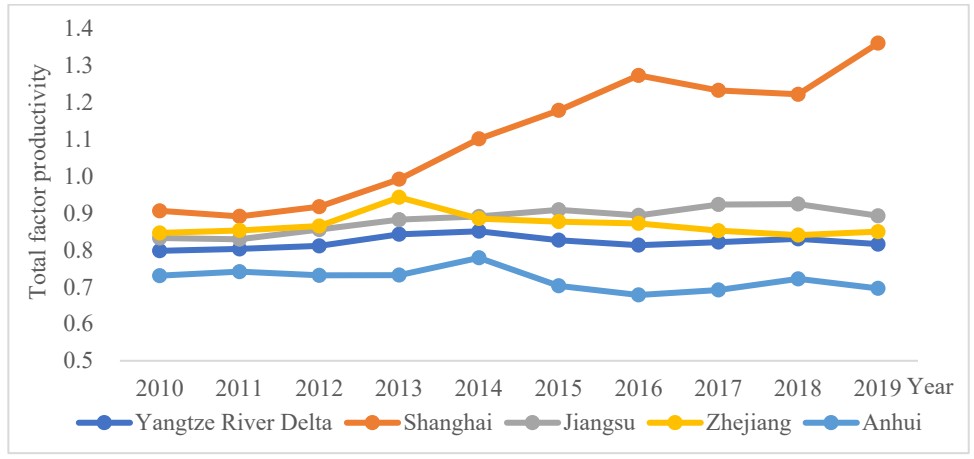

**Figure 3.** Calculated value of total factor productivity of cities in the Yangtze River Delta from 2010 to 2019.

Figure 4 shows the dynamic evolution of TFP in 41 cities in the Yangtze River Delta in 2010, 2015, and 2019 and the average value in 2010–2019. In general, the TFP level of these cities was high; however, the spatial differences were significant. The high-value areas were mainly distributed in the coastal areas and along the Yangtze River. The Shanghai metropolitan area has become the leader of regional economic development. The low-value areas were mainly distributed in the northern part of the Yangtze River Delta and the western and southern parts of Anhui. Specifically, in 2010, there were 4 cities with TFPs exceeding 1, namely, Suzhou (1.1793), Taizhou (1.1028), Jinhua (1.0037), and Maanshan (1.0702), among which Suzhou was the highest; in 2015, there were 6 cities with TFPs exceeding 1, namely, Shanghai (1.1791), Suzhou (1.0942), Zhenjiang (1.1064), Jiaxing (1.1046), Jinhua (1.2102), and Tongling (1.0964), among which Jinhua was the highest, followed by Shanghai; and in 2019, the number of cities with TFPs exceeding 1 increased to 8 cities, namely, Shanghai (1.3616), Wuxi (1.0856), Suzhou (1.1352), Nantong (1.0606), Jiaxing (1.0004), Jinhua (1.0492), Lishui (1.0241), and Bengbu (1.1338), among which Shanghai was the highest at 1.362. According to the average TFPs of cities in the Yangtze River Delta from 2010 to 2019, it can be observed that cities with better economic development are mainly distributed along the coast and along the Yangtze River. The cities with TFPs exceeding 1 were Shanghai, Suzhou, Taizhou, Jiaxing, and Jinhua; Suzhou was the highest (1.133), followed by Shanghai (1.108). Cities with relatively poor economic development were

mainly distributed in the northern part of the Yangtze River Delta and the western and southern parts of Anhui. Huangshan, Huaibei, Xuancheng, Lu'an, Chizhou, and Chuzhou have become economic development depressions in the Yangtze River Delta region, with an average TFP of about 0.6.

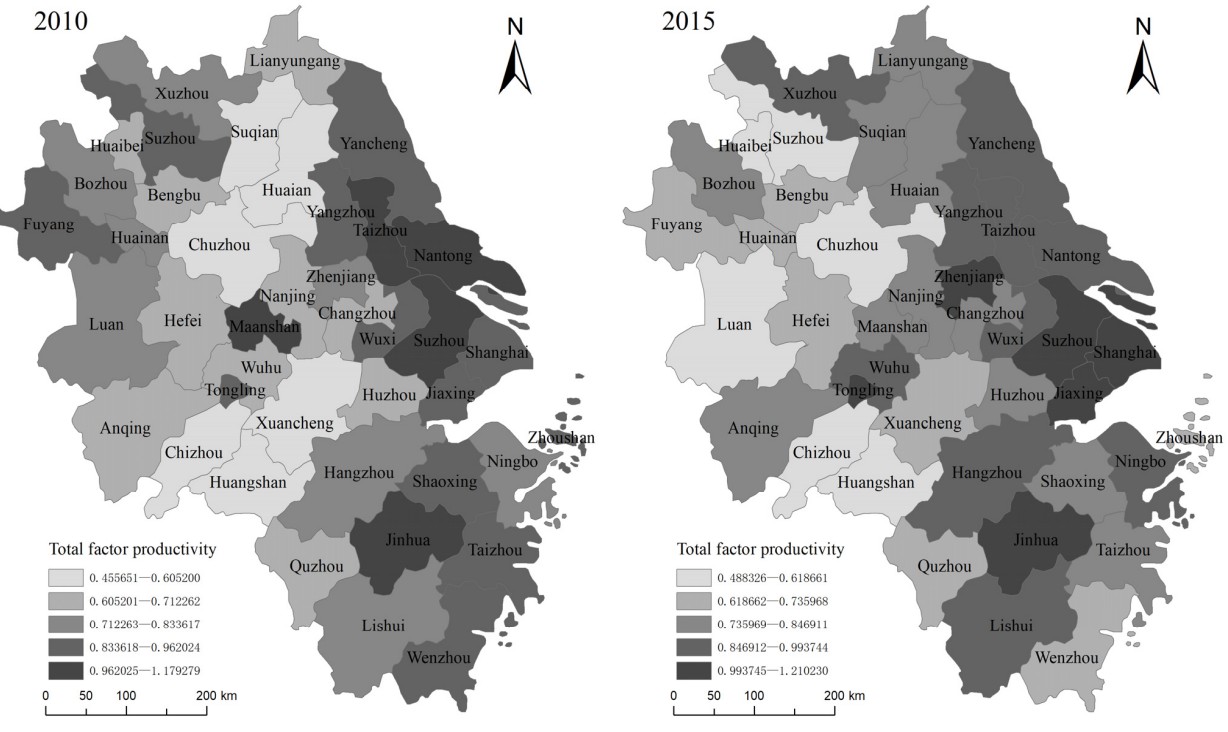

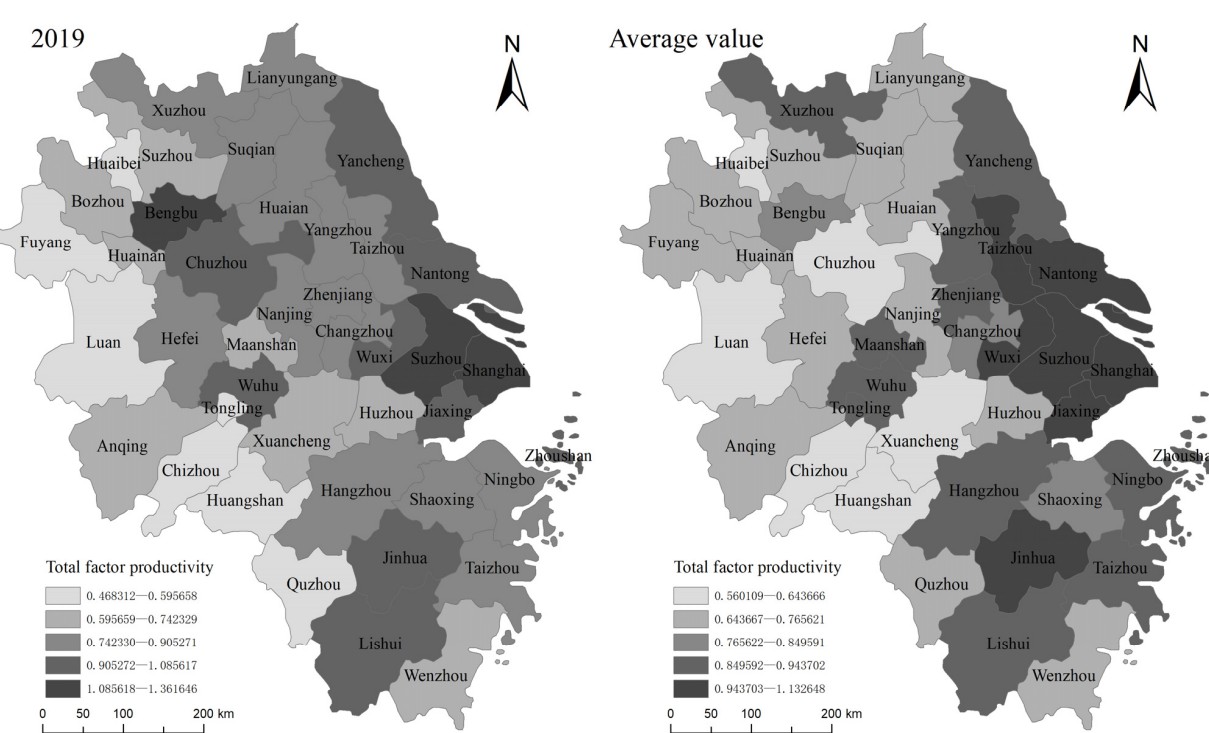

**Figure 4.** The spatial dynamic evolution of total factor productivity in cities in the Yangtze River Delta from 2010 to 2019.

## 5. Influence Mechanism

### 5.1. Model Construction and Variable Selection

We constructed the full-sample panel data of 41 cities in the Yangtze River Delta from 2010 to 2019 and analyzed the impact of intra-city and inter-city innovation networks on urban economic growth. Compared to the traditional research using regional economic aggregates to measure city economic growth, calculating the TFP based on input and output variables can more comprehensively reflect the level of urban economic growth. We used the urban TFP as a dependent variable, which represents urban economic growth, while the core independent variables included intra-city and inter-city innovation networks, measured by the scale of intra-city and inter-city cooperative invention patents, respectively. The data were sourced from the incoPat patent database.

Referring to the relevant studies [36], we selected control variables that had an important impact on urban economic growth, including economic development level, degree of openness, city size, population density, and fiscal autonomy of local governments. To reduce the effect of heteroskedasticity, we logged all variables. Table 3 shows the descriptive statistics of the variables.

**Table 3.** Descriptive statistics of related variables.

| Variables | Symbol | Description | Mean | SD | Minimum | Maximum |
|---|---|---|---|---|---|---|
| Total factor productivity | TFP | Super-efficient total factor productivity | 0.821 | 0.181 | 0.421 | 1.865 |
| Inter-city cooperation | Inter | Number of cooperative patents between cities (piece) | 421.82 | 825.55 | 0 | 5463 |
| Intra-city cooperation | Intra | Number of cooperative patents in the city (piece) | 394.55 | 1010.1 | 0 | 7191 |
| Economic development | GDP | GDP per capita (CNY/person) | 67,051.6 | 32,818.7 | 9068 | 199,017 |
| Degree of openness | FDI | Foreign direct investment/GDP (%) | 0.463 | 0.368 | 0.029 | 2.853 |
| City size | Size | Population of the city (10,000 people) | 207.76 | 225.40 | 29 | 1469 |
| Population density | Dens | Population of the city/urban area (10,000 people/square kilometer) | 1.2762 | 0.5983 | 0.5630 | 3.8535 |
| Financial autonomy | GOV | Public budget revenue/public budget expenditure (%) | 0.641 | 0.232 | 0.069 | 1.116 |

The model was constructed as follows:

$$\text{LnTFP}_{ct} = \alpha + \beta_1 \text{LnInter}_{ct} + \beta_2 \text{LnIntra}_{ct} + \beta_3 \text{LnInter}_{ct} * \text{LnIntra}_{ct}$$
$$+ \beta_4 \text{LnInter}^2_{ct} + \beta_5 \text{LnIntra}^2_{ct} + \beta_6 \text{LnGDP}_{ct} + \beta_7 \text{LnFDI}_{ct}$$
$$+ \beta_8 \text{LnSize}_{ct} + \beta_9 \text{LnDens}_{ct} + \beta_{10} \text{LnGOV}_{ct} + \mu_c + \varepsilon$$

where $c$ and $t$ represent the city and year, respectively; $\text{LnTFP}_{ct}$ is the logarithm of the TFP of city $c$ in year $t$; $\text{LnIntra}_{ct}$ and $\text{LnInter}_{ct}$ are the logarithms of the intra-city and inter-city innovation network scales, respectively, in city $c$ in year $t$; $\text{LnIntra}^2_{ct}$, $\text{LnInter}^2_{ct}$, $\text{LnInter}_{ct} * \text{LnIntra}_{ct}$ represent the logarithmic forms of the square and interaction terms of the intra-city and inter-city innovation network scales in city $c$ year $t$; $\text{LnGDP}_{ct}$, $\text{LnFDI}_{ct}$, $\text{LnSize}_{ct}$, $\text{LnDens}_{ct}$, $\text{LnGOV}_{ct}$, respectively, represent the logarithmic form of the control variable in $c$ city $t$ year; $\mu_c$ represents the fixed city effect; and $\alpha$ and $\varepsilon$ represent the constant and random-error terms.

### 5.2. Analysis of Model Regression Results

Using the full sample data of the Yangtze River Delta urban innovation networks from 2010 to 2019 to draw a scatter diagram of the intra-city and inter-city innovation networks (Figure 5), we found that there was a high correlation between the two, which also confirmed the above conclusion on their spatial structure similarities. At the same time, using Stata software to conduct a correlation analysis, we found that the correlation coefficient between the two was as high as 0.8437. In order to avoid the model being

affected by the correlation between the independent variables, the regression analysis of the influence of intra-city and inter-city innovation networks on regional TFP was conducted.

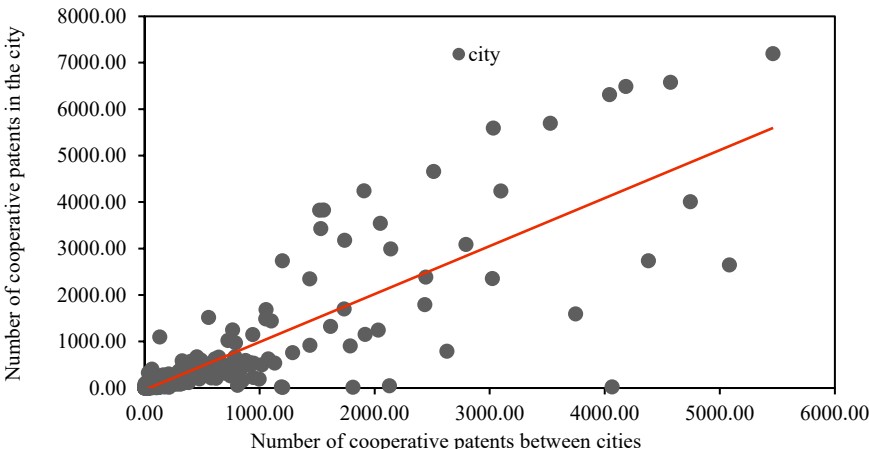

**Figure 5.** Scatter diagram of the distribution of the intra-city and inter-city innovation networks in the Yangtze River Delta from 2010 to 2019.

The results of multiple regression using panel data are shown in Table 4. Model 1 mainly depicts the influence of control variables on urban TFPs. The economic development level index is significantly positive, indicating that cities with higher economic development levels have higher TFPs. The index of openness is significantly negative, indicating that foreign direct investment (FDI) has not significantly promoted the economic growth of cities in the Yangtze River Delta. This may be because the FDI introduced by China may be mainly concentrated in low-tech fields, and not only did this not produce a high technology spillover effect, it also produced a strong crowding-out effect. The index of city size is significantly negative, indicating that cities in the Yangtze River Delta have diseconomies of scale; as cities expand, large-scale urban diseases, such as traffic congestion and pollution, cause external diseconomies, and smaller cities cannot obtain economies of scale. The index of population density is significantly positive, indicating that the agglomeration economy brought about by labor agglomerations in these cities has a positive effect on urban development, which further verifies the relevant research conclusions of Huang and He using population density to measure the agglomeration economy [37]. The index of financial autonomy is positive, but does not reach statistical significance, indicating that the political and economic development strategies adopted by local governments have a stable impact on urban economic efficiency; in general, the more active the local government's fiscal policy, the more likely it is for the city's economic efficiency to increase.

The core independent variables of models 2 and 3 are the primary terms of the scales of inter-city and intra-city innovation networks, respectively, and the core independent variables of models 4 and 5 are the secondary terms of their scales, respectively. It can be seen from the results that the first-order coefficient of the inter-city innovation networks is positive; however, it does not reach statistical significance; after adding the square term of the scale of the inter-city innovation networks, the coefficient of the square term is significantly positive, indicating that the inter-city innovation network has a non-linear positive impact on urban economic growth. There may be a positive U-shaped relationship between inter-city innovation networks and urban economic growth.

In the low-level stage of inter-city innovation networks, the positive effect on urban economic growth gradually diminishes, because most of the cities in the low-level stage are underdeveloped. The attractiveness and absorptive capacity of these cities for non-local knowledge are at a low level [38]; therefore, it is difficult to build high-level inter-city innovation networks, and its role in promoting urban economic growth is also weak. Conversely, at the high-level stage, inter-city innovation networks can significantly promote urban economic growth, and the larger the scale is, the stronger the promotion effect. This

is because core cities have a large number of heterogeneous innovation actors, and they are widely used in a broad range of the economic and technological fields conducting innovation activities and, thus, their demands, attractiveness, and absorptive capacity for non-local knowledge are high. Therefore, most of the inter-city innovation networks constructed are of high quality and have better innovation effects, which can bring vitality to urban economic development and promote growth.

**Table 4.** Regression results of urban innovation networks and regional growth in the Yangtze River Delta urban agglomeration from 2010 to 2019.

| | Model (1) | Model (2) | Model (3) | Model (4) | Model (5) | Model (6) |
|---|---|---|---|---|---|---|
| | LnTFP | LnTFP | LnTFP | LnTFP | LnTFP | LnTFP |
| LnInter | | 0.0294 | | −0.314 ** | | −0.0760 * |
| | | (0.0434) | | (0.131) | | (0.0443) |
| LnIntra | | | −0.0628 | | −0.0488 | −0.278 *** |
| | | | (0.0409) | | (0.124) | (0.0749) |
| LnInter*LnIntra | | | | | | 0.471 *** |
| | | | | | | (0.11) |
| LnInter^2 | | | | 0.478 *** | | |
| | | | | (0.174) | | |
| LnIntra^2 | | | | | −0.0231 | |
| | | | | | (0.162) | |
| LnGDP | 0.0763 *** | 0.0756 *** | 0.0790 *** | 0.0796 *** | 0.0789 *** | 0.0803 *** |
| | (0.0171) | (0.0172) | (0.0178) | (0.0174) | (0.0179) | (0.0179) |
| LnFDI | −0.0243 *** | −0.0238 *** | −0.0249 *** | −0.0247 *** | −0.0250 *** | −0.0244 *** |
| | (0.00637) | (0.00642) | (0.00646) | (0.00639) | (0.00649) | (0.00649) |
| LnSize | −0.122 *** | −0.124 *** | −0.122 *** | −0.108 *** | −0.122 *** | −0.116 *** |
| | (0.0282) | (0.0282) | (0.0283) | (0.0297) | (0.0285) | (0.0287) |
| LnGOV | 0.0171 | 0.0169 | 0.0168 | 0.0125 | 0.017 | 0.0143 |
| | (0.0179) | (0.0181) | (0.0178) | (0.0201) | (0.0178) | (0.0191) |
| LnDens | 0.145 *** | 0.146 *** | 0.146 *** | 0.133 *** | 0.146 *** | 0.144 *** |
| | (0.04) | (0.0398) | (0.0404) | (0.0402) | (0.0407) | (0.0402) |
| constant | 0.573 *** | 0.582 *** | 0.580 *** | 0.443 *** | 0.583 *** | 0.536 *** |
| | (0.142) | (0.144) | (0.141) | (0.161) | (0.144) | (0.144) |
| Region fixed effect | Yes | Yes | Yes | Yes | Yes | Yes |
| Observations | 410 | 410 | 410 | 410 | 410 | 410 |
| R-squared | 0.688 | 0.688 | 0.688 | 0.694 | 0.688 | 0.695 |

* $p < 0.10$; ** $p < 0.05$; *** $p < 0.01$.

For intra-city innovation networks, the coefficients of the primary and quadratic terms are both negative and do not reach statistical significance, indicating that, according to the existing data, these networks have no significant impact on the urban economic growth in the Yangtze River Delta, while the quadratic term is negative, which indicates that there may be an inverted U-shaped relationship between intra-city innovation networks and urban economic growth. Intra-city innovation networks represented by cooperative patents play a certain role in promoting regional economic development; however, too many of them may bring about a redundancy of knowledge, which leads to the development dilemma of path locking in the region.

Model 6 takes the interaction term of intra-city and inter-city innovation networks as the core independent variable, exploring the mechanism of interaction between them and urban economic growth. The results show that the interaction term is significantly positive, indicating that they have complementary roles in promoting growth. Studies have shown that cities with dense local and non-local innovation networks tend to achieve better development performances [39]; the dense local network creates a good innovation environment and also strengthens the region's ability to absorb non-local knowledge to transform it and apply it locally [40]. At the same time, inter-city innovation networks provide innovation vitality and expand external knowledge sources for the development

of intra-city innovation networks, thus avoiding industrial and technological stagnations or lock-ins due to locally contained interactions and over-embedding within a regionally rigid inward-looking system [41]. The interaction of intra-city and inter-city innovation networks provides an inexhaustible impetus for urban economic growth.

To verify the robustness of the benchmark model, a robustness test was performed with the urban TFP lagging one period as the dependent variable, and the results are shown in Table 5. From the regression results, the coefficients and symbols of the core explanatory variables are consistent with the benchmark regression results in Table 4, which further verifies the robustness of the model results presented in Table 4.

**Table 5.** Robustness test of the regression results of urban innovation networks and regional growth in the Yangtze River Delta urban agglomeration from 2010 to 2019.

|  | Model (7) | Model (8) | Model (9) | Model (10) | Model (11) |
|---|---|---|---|---|---|
|  | L.LnTFP | L.LnTFP | L.LnTFP | L.LnTFP | L.LnTFP |
| LnInter | 0.0542 |  | −0.310 ** |  | −0.0453 |
|  | (0.0469) |  | (0.137) |  | (0.0521) |
| LnIntra |  | −0.0533 |  | −0.0186 | −0.253 ** |
|  |  | (0.0523) |  | (0.164) | (0.0894) |
| LnInter*LnIntra |  |  |  |  | 0.420 *** |
|  |  |  |  |  | (0.123) |
| LnInter^2 |  |  | 0.505 *** |  |  |
|  |  |  | (0.171) |  |  |
| LnIntra^2 |  |  |  | −0.0581 |  |
|  |  |  |  | (0.217) |  |
| Control variables | Yes | Yes | Yes | Yes | Yes |
| constant | 0.538 *** | 0.514 ** | 0.371 | 0.524 ** | 0.497 * |
|  | (0.218) | (0.215) | (0.230) | (0.213) | (0.218) |
| Region fixed effect | Yes | Yes | Yes | Yes | Yes |
| Observations | 369 | 369 | 369 | 369 | 369 |
| R-squared | 0.684 | 0.684 | 0.690 | 0.684 | 0.690 |

* $p < 0.10$; ** $p < 0.05$; *** $p < 0.01$.

To sum up, the empirical analysis shows that there are different mechanisms of action between intra-city and inter-city innovation networks and urban economic growth. The relevant mechanisms are summarized in Figure 6.

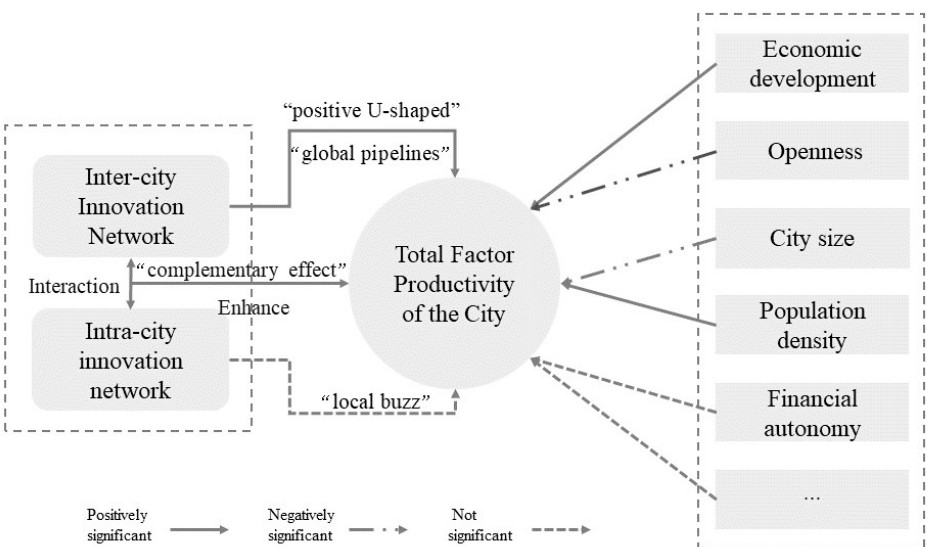

**Figure 6.** Diagram of the interaction mechanism between urban innovation networks and regional growth in the Yangtze River Delta.

## 6. Discussion and Conclusions

Based on the panel data of cooperative application for invention patents and the socio-economic statistics of cities in the Yangtze River Delta from 2010 to 2019, the input–output index system of economic growth was established in this study. With the methods of TFP and multiple regression analysis, this paper explored the dynamic characteristics, spatial evolution, and influencing mechanism of intra-city and inter-city innovation networks on economic growth in the Yangtze River Delta.

First, the connection density of intra-city and inter-city innovation networks in this region increased, and the spatial structure presented a highly similar core–edge structure. Specifically, the Z-shaped spatial structure of the networks with Hefei–Nanjing–Shanghai–Hangzhou–Ningbo as the core became clearer. Comparatively speaking, most of Anhui, western Zhejiang, and northern Jiangsu became the edge of the innovation networks in the Yangtze River Delta. This spatial distribution was in agreement with the distribution of the economic structure and innovation ability among cities in this area. Due to the implementation of policies, such as regional integration and industrial transfer, the intra-city and inter-city innovation networks presented a trend of core-driven marginal development towards the overall regional coordinated development.

Second, the economic growth of the 41 cities in the Yangtze River Delta was generally in a relatively stable state, and few cities were characterized by fluctuations. Specifically, the level of economic growth in Shanghai was the highest and increasing, followed by Jiangsu and Zhejiang, while Anhui became the depression of economic development in the region. From the perspective of spatial distribution, unlike the Z-shaped core–edge structure of intra-city and inter-city innovation networks, the core cities were distributed along the coast and along the golden waterway of the Yangtze River; the metropolitan area with Shanghai as the core became the leader of regional economic development.

Third, there was strong heterogeneity in the influence of innovation networks on economic growth. This influence presented a positive non-linear relationship and occurred when the inter-city innovation networks reached a certain scale, which could significantly promote city economic growth; however, the influence was not significant, but the interaction between them was significantly positive, which indicated that intra-city and inter-city innovation networks had a strong complementary effect on promoting economic growth. Intensive intra-city cooperation can acquire local tacit knowledge that is difficult for inter-city networks to acquire, while the inter-city collaboration can acquire external innovation knowledge that is difficult for intra-city innovation networks to acquire.

Theoretically, there was heterogeneity in the action mechanism of the networks based on cooperative invention patents on economic growth. We suggested strengthening the research on the influence mechanisms, such as network capital-oriented, social capital-oriented, formal, and informal, to provide a theoretical reference for analyzing the source of economic growth from the perspective of innovation networks. Practically, our suggestions are as follows: first, adhere to the six-in-one policy of government and industry-university research to promote the cooperation of regional science and technology innovation in the Yangtze River Delta, and provide a power source for the science and technology innovation community and regional high-quality integration; second, insist on performing good industrial transfer in central cities and undertaking industrial transfers in marginal areas, and promote the coordinated development of marginal areas with industrial transfer and industrial cooperation in core cities; third, to support the construction of intra-city and inter-city innovation networks, such as industrial technology innovation alliances, innovation enclaves, industrial technology transfer, and transformation platforms, to enhance inter-regional knowledge flow, regional innovation capabilities, and promote regional economic development.

**Author Contributions:** Conceptualization, X.C. and Z.Y.; methodology, B.C. and Y.G.; software, B.C.; validation, B.C., Y.G. and X.C.; formal analysis, X.C.; data curation, B.C.; writing—original draft preparation, X.C. and B.C.; writing—review and editing, X.C. and Z.Y.; visualization, Y.G.; project administration, X.C.; funding acquisition, X.C. All authors have read and agreed to the published version of the manuscript.

**Funding:** This research was funded by National Natural Science Foundation of China (42171184 and 42130510).

**Data Availability Statement:** The data presented in this study are available on request from the corresponding author.

**Conflicts of Interest:** The authors declare no conflict of interest.

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
