# Peer review of "The Impact of Intra-City and Inter-City Innovation Networks on City Economic Growth: A Case Study of the Yangtze River Delta in China"

_land, doi:10.3390/land12071463_

Round 1

Reviewer 1 Report (Previous Reviewer 2)

The manuscript is now suitable for publication.

Author Response

Reviewer 1

Comments and Suggestions for Authors

The manuscript is now suitable for publication.

Reply: Thanks very much.

Reviewer 2 Report (New Reviewer)

My thoughts and recommendations are listed below.

(1) In general, I believe that this article examined the link between the innovation network and economic expansion. Additionally, the verification was carried out using a variety of quantitative model methodologies.

(2) The calculation of innovation networks, I'm afraid, is one of my worries. According to the paper, "we use the number of collaborations between innovation entities to measure the intensity of collaborative innovation within the city." However, in my opinion, the article spends more time introducing the methodology than it does elaborating on this crucial information in the following sections. In addition, I fear that it will be difficult to both define "innovation entities" and gather this information.

(3) Regarding "the patent data," another crucial factor. We can extract patent data for a specific city from the statistical yearbook, however I don't understand the network these nodes build.

(4) The model is initially constructed by applying the C-D production function, however the model dependent variables between them have entirely distinct meanings. 

Nothing

Author Response

Reviewer 2

Comments and Suggestions for Authors

My thoughts and recommendations are listed below.

(1) In general, I believe that this article examined the link between the innovation network and economic expansion. Additionally, the verification was carried out using a variety of quantitative model methodologies.

Reply: Thanks very much. Using the patent data of cooperative inventions and the panel data of socio-economic statistics for 2010-2019, this study quantitatively analysed the spatial structure evolution of intra-city and inter-city innovation networks for 41 cities in the Yangtze River Delta and their influence on economic growth.

(2) The calculation of innovation networks, I'm afraid, is one of my worries. According to the paper, "we use the number of collaborations between innovation entities to measure the intensity of collaborative innovation within the city." However, in my opinion, the article spends more time introducing the methodology than it does elaborating on this crucial information in the following sections. In addition, I fear that it will be difficult to both define "innovation entities" and gather this information.

Reply: Thanks very much. This may be a mistake in our study, "innovation entities" should be “innovation actors”, including government, enterprises, universities, research institutions and intermediary service agencies. We have revised them.

(3) Regarding "the patent data," another crucial factor. We can extract patent data for a specific city from the statistical yearbook, however I don't understand the network these nodes build.

Reply: Thanks very much. A joint patent application is an interactive innovation process based on knowledge shar-ing between organisations and resource integration embedded in social networks. Eco-nomic geographers have widely recognised that joint patent application data can be re-searched on innovation networks and knowledge spillovers The patent data is sourced from the incoPat database and processed as follows: (i) Time selection. Because it takes 18 months for patent applications to be published, our research limited the patent application dates to the period from January 1, 2010 to December 31, 2019, and the locations to Shanghai, Anhui, Zhejiang and Jiangsu Province. The patents with 1 number of applicants were filtered, and individuals were filtered in the applicant category to obtain 256,934 joint patent applications in the Yangtze River Delta; (ii) Match-ing the innovation actors and geographical location. We matched the geographical posi-tion of the applying entitiesactors by using the enterprise database of Tianyanchao (https://www.tianyancha.com/search), and cross-processed the data of three or more ac-tors, thus obtaining 400,201 linkages of city innovation networks; (iii) The classification of innovation networks. We divided the innovation networks within the Yangtze River Delta into intra-city and inter-city ones, and finally obtained 161,766 linkages of intra-city and 86,487 linkages of inter-city innovation networks

(4) The model is initially constructed by applying the C-D production function, however the model dependent variables between them have entirely distinct meanings.

Reply: Thanks very much. Based on the idea of C-D production function, we set up the calculation method of TFP, which is also the method commonly used in the existing research. This paper studies the TFP of cities as a decision-making unit, which is based on the (Cobb- Douglas, C-D) production function model in economic growth theory. The essence of national and regional economic development is the input of capital, land, labour and other production factors. According to the neoclassical economic growth theory and the understanding of economic growth in cities and urban agglomerations, this study selected capital, land and labour as input indicators and urban economic aggregates as output indicators.

Reviewer 3 Report (New Reviewer)

Congratulations on a terrific study on intra- and inter-city networks!

Moderate editing of English is required.

Author Response

Reviewer 3

Comments and Suggestions for Authors

Congratulations on a terrific study on intra- and inter-city networks!

Comments on the Quality of English Language

Moderate editing of English is required.

Reply: Thanks very much. We have polished the English of the article.

This manuscript is a resubmission of an earlier submission. The following is a list of the peer review reports and author responses from that submission.

Round 1

Reviewer 1 Report

The paper examines the evolution of intra-city and inter-city innovation networks in the Yangtze River Delta and their impact on economic growth. While the topic is interesting, there are some areas where improvements could be made to strengthen the paper and derive more policy conclusions. Here are my recommendations:

1.      The introduction could be more concise and avoid repetition. Specifically, the first and third paragraphs need reorganizing.

2.     The literature review section needs to be more comprehensive and closely related to the research gap. Section 2.1 requires more references to support its claims.

3.      The method used to construct the intra-city and inter-city innovation networks should be described more transparently and in detail.

4.     Ensure that all figures and tables are of high resolution to support analysis and conclusions, including Figure 2, which is difficult to interpret due to low resolution.

5.      The author's characterization of the intra-city and inter-city innovation networks as having a core-periphery structure is subjective and would benefit from a more rigorous analysis. The author should refer to Rombach et al. (2014) and Zhang et al. (2019) for measures of network structure.

Rombach et al. (2014). Core-periphery structure in networks.

Zhang et al. (2019). Mesoscale structures in world city networks.

6.     In section 5, it is unclear why you emphasize the impact of innovation networks on economic growth, as the model shows that the number of patent collaborations within and between cities has an impact on economic growth. Therefore, even if the title is changed to “The impact of patent collaborations within and between cities on economic growth,” it would still be appropriate. So, what is your reason for emphasizing the impact of innovation networks in this paper?

7.      In the model, it is not clear why the quadratic term and interaction terms of the core explanatory variables are included. The author should provide a rationale for their inclusion.

The English writing needs further revision and refinement to meet academic standards. For example, Page 2 Line 72-73: “Third, what is the impact of intra-city and inter-city innovation networks on economic growth in the Yangtze River Delta and what is the impact mechanism?”

Author Response

Dear,

Thank you very much for the comments of the reviewers. We have carefully revised them. The following is our response:

Reviewer 1 Comments:

The paper examines the evolution of intra-city and inter-city innovation networks in the Yangtze River Delta and their impact on economic growth. While the topic is interesting, there are some areas where improvements could be made to strengthen the paper and derive more policy conclusions. Here are my recommendations:

  1. The introduction could be more concise and avoid repetition. Specifically, the first and third paragraphs need reorganizing.

Reply: Thanks very much. We have revised the introduction.

“1. Introduction

Economic geographers have long paid attention to the hot topics of innovation, innovation network and economic growth (Marques and Morgan, 2021). Innovation network refers to the cooperation of innovation actors such as government, enterprises, universities, research institutions and intermediary service agencies in technology re-search and development (Gluckler, 2014). Considering cities as boundaries, innovation networks can be divided into intra-city and inter-city ones. With the economic global-isation, the innovation paradigm has changed from the traditional closed linear model to the modern open network one, and the influence of innovation networks on regional innovation and economic growth have gradually become the frontier scientific prob-lem for economic geographers (Fernandes et al., 2020; Bathelt et al., 2018). As the rep-resentative of developed regions in developing countries, The region of Yangtze River Delta has a rapid economic development and a strong innovation atmosphere, and the Chinese Central government attaches great importance to the construction of a regional innovation community. On December 20th, 2020, the Ministry of Science and Tech-nology in China released the Development Plan for the Construction of Innovation Community in the Yangtze River Delta, which clearly pointed out that it was necessary to focus on high-tech industries such as biomedicine, new materials, integrated circuits, equipment manufacturing, and to achieve cross-border cooperation with innovation actors such as universities, research institutions, enterprises and intermediaries in 41 cites, so as to promote the free flow of innovation elements. All these make it a typical region for studying intra-city and inter-city innovation networks; however, little re-search has focused on the relationship between innovation network and economic growth, especially in terms of the former influencing the latter.

Economic geographers have studied the structure characteristics, spatial scale, in-fluence mechanism and effect of innovation networks, among them, the research on innovation networks in different spatial scales, such as global, local, and global-local, has attracted the attention of many scholars; however, such research generally focuses on the analysis of network structure rather than on relationship mechanisms, and the influence of innovation networks of different spatial scales (intra-city and inter-city) on economic growth remains controversial (Denney et al., 2021). Therefore, taking 41 cit-ies in the Yangtze River Delta as examples, this paper analyses the spatial and temporal evolution characteristics of intra-city and inter-city innovation networks and economic growth, trying to answer the following three questions: first, what are the characteris-tics of intra-city and inter-city innovation networks in the Yangtze River Delta and the differences between them? Second, what are the characteristics of spatial and temporal evolution of economic growth in the Yangtze River Delta? Third, what is the impact of intra-city and inter-city innovation networks on economic growth in the Yangtze River Delta and what is the impact mechanism?

The remainder of this paper is organised as follows: the next section presents the literature review. Section 3 describes the data and empirical variables used in analysing the innovation networks and economic growth. With the help of ArcGIS 10.6 and MaxDEA, section 4 analyses the spatio-temporal evolution of innovation networks and economic growth in the Yangtze River Delta. Based on regression model, section 5 discusses the mechanism of innovation networks impact on economic growth. The last section concludes and discusses the paper.”

  1. The literature review section needs to be more comprehensive and closely related to the research gap. Section 2.1 requires more references to support its claims.

Reply: Thanks very much. We have revised the literature.

“2. Literature Review

2.1. Relationship between innovation networks and economic growth

New knowledge based on technological innovation is recognised by scholars as the foundation of promoting economic growth (Antonelli et al., 2011; Mahmood et al., 2022). On the basis of traditional economic growth theory, endogenous economic growth theory emphasises the key role of knowledge in driving productivity and economic growth (Aghion and Howitt, 1998). It also points out that economic growth and innovation networks are interrelated, as knowledge creation, accumulation and transfer can effectively explain the differences in city economic growth levels (Roberts and Setterfield, 2010). Innovation network is an important way to promote regional growth (Capello et al., 2018). The purpose of innovation network is to improve the innovation ability and performance of innovation actors, and then to transform the research and development of innovative products into economic benefits. Innovation can be regarded as one of the necessary conditions for regional economic growth and development. The existing research has also shown that the knowledge flow in inno-vation network determines the technological innovation ability and the level of re-gional economic griwth (Ramadani et al., 2013).

Innovation networks strengthen the knowledge flow inside and outside the region, and is a key capital investment in the process of regional economic growth(Huggins and Thompson, 2017). However, there is still controversy about the relationship be-tween innovation networks of different spatial scales and economic growth. Local knowledge creation and global knowledge acquisition interact with regional economic growth (Esposito and Rigby, 2019), and we can see knowledge integrators with high competitiveness innovate by integrating global and local knowledge (Buciuni and Pi-sano, 2018). Crespo (2014) mainly analysed the influence of the structural attributes of local knowledge networks on the promotion of regional competitiveness, while Breschi (2017) pointed out that non-local innovation networks are more conducive to regional economic development. There are great differences in the influence of different spatial scales of knowledge on regional development; the course of regional economy and innovation not only depends on localised production and knowledge creation, but also needs to combine the “local buzz” and “global pipelines” (Bathelt et al., 2004; Storper, 2018; Galaso & KováÅ™ík, 2021). At the same time, it should be emphasized that there are costs for innovation subjects to cooperate with local and non-local innovators. Es-posito et al. (2019) found that too much local interaction would lead to the disap-pearance of the boundaries of innovation actors and the decline of regional techno-logical innovation, when innovation subjects engage in non-local interaction, the costs may also exceed the benefits. Bianchi et al. (2021) find that acting as interregional broker cities, especially connecting Latin American cities to the rest of the world, neg-atively affects patent outcomes.”

  1. The method used to construct the intra-city and inter-city innovation networks should be described more transparently and in detail.

Reply: Thanks very much. The main steps in screening and processing the data were as follows: (i) because it takes 18 months for patent applications to be published, our research limited the patent application dates to the period from January 1, 2010 to December 31, 2019, and the locations to Shanghai, Anhui, Zhejiang and Jiangsu Province. The patents with 1 number of applicants were filtered, and individuals were filtered in the applicant cat-egory to obtain 256,934 joint patent applications in the Yangtze River Delta; (ii) we matched the geographical position of the applying entities by using the enterprise da-tabase of Tianyanchao (https://www.tianyancha.com/search), and cross-processed the data of three or more actors, thus obtaining 400,201 linkages of city innovation net-works; (iii) we divided the innovation networks within the Yangtze River Delta into intra-city and inter-city ones, and finally obtained 161,766 linkages of intra-city and 86,487 linkages of inter-city innovation networks

  1. Ensure that all figures and tables are of high resolution to support analysis and conclusions, including Figure 2, which is difficult to interpret due to low resolution.

Reply: Thanks very much. We have redrawn the figures and tables.

  1. The author's characterization of the intra-city and inter-city innovation networks as having a core-periphery structure is subjective and would benefit from a more rigorous analysis. The author should refer to Rombach et al. (2014) and Zhang et al. (2019) for measures of network structure.

Rombach et al. (2014). Core-periphery structure in networks.

Zhang et al. (2019). Mesoscale structures in world city networks.

Reply: Thanks very much. The intra-city innovation networks of the Yangtze River Delta present a "core-periphery" structure, and the core changed from a single centre in Shanghai to multi-centres "Nanjing-Shanghai-Hangzhou-Ningbo". This result also accords with Rombach et al. (2014) and Zhang et al. (2019), and refers to their measurement of network structure. Specifically, the highlands of intra-city innovation cooperation are mainly concentrated in the core cities of Shanghai, Jiangsu and Zhejiang. Anhui, southwest of Zhejiang and north of Jiangsu has become depressions for intra-city co-operation. In 2010, Shanghai became the single core of the Yangtze River Delta with 2,736 intra-city innovation collaborations. Hangzhou was far behind, with only 902 intra-city innovation collaborations; the difference between the two is more than three times. In 2019, Shanghai and Nanjing became the core with 7191 and, respectively, 6488 intra-city innovation collaborations. In addition, Hangzhou, Ningbo and Suzhou each had 4,007, 3,176, and 3,088 intra-city innovation collaborations as the sub-core. Only Hefei in Anhui entered the top ten with 1,686 intra-city collaborations. From the perspective of growth and changes, from 2010 to 2019, intra-city innovation coopera-tion in 41 cities in the Yangtze River Delta was on the rise, the core cities maintaining a growth rate of 3 to 10 times; Nanjing had seen an increase of nearly 10 times in the past 10 years, totalling 5,915 intra-city innovative collaborations, while the peripheral cities, such as Bozhou, Fuyang, Suqian or Lishui, due to the relatively small base of innovation cooperation within the city in 2010, Significant increase in 10 years, how-ever there is still a big gap between the absolute amount of growth and the core cities.

  1. In section 5, it is unclear why you emphasize the impact of innovation networks on economic growth, as the model shows that the number of patent collaborations within and between cities has an impact on economic growth. Therefore, even if the title is changed to “The impact of patent collaborations within and between cities on economic growth,” it would still be appropriate. So, what is your reason for emphasizing the impact of innovation networks in this paper?

Reply: Thanks very much. On the basis of traditional economic growth theory, endogenous economic growth theory emphasises the key role of knowledge in driving productivity and economic growth (Aghion and Howitt, 1998). It also points out that economic growth and innovation networks are interrelated, as knowledge creation, accumulation and transfer can effectively explain the differences in city economic growth levels (Roberts and Setterfield, 2010). Innovation network is an important way to promote regional growth (Capello et al., 2018). The purpose of innovation network is to improve the innovation ability and performance of innovation actors, and then to transform the research and development of innovative products into economic benefits. Innovation can be regarded as one of the necessary conditions for regional economic growth and development. The existing research has also shown that the knowledge flow in inno-vation network determines the technological innovation ability and the level of re-gional economic griwth (Ramadani et al., 2013).Innovation networks strengthen the knowledge flow inside and outside the region, and is a key capital investment in the process of regional economic growth(Huggins and Thompson, 2017). So, we emphasize the impact of innovation networks in this paper.

Patent literature is the largest source of technical information in the world. The report of the World Intellectual Property Organization (WIPO) shows that about 90-95 percent of global R&D outputs is contained in patents, and the rest is embodied in scientific literature, such as papers and publications (Prabhakaran et al., 2015). Patent literature offers the advantages of openness, timeliness, detailed content, and easy comparison between industrial technologies or different spaces, and has become an important data source for studying knowledge production and innovation activities (Ter Wal, 2013). A joint patent application is an interactive innovation process based on knowledge sharing between organisations and resource integration embedded in so-cial networks. Economic geographers have widely recognised that joint patent appli-cation data can be researched on innovation networks and knowledge spillovers (Patra and Muchie, 2022), the invention patent represents the original technology, which can better reflect the technological innovation achievements. In this study, the joint patent application data for the target region were selected to depict the intra-city and in-ter-city innovation networks and were sourced from the incoPat database (https://www.incopat.com/). So, the number of patent collaborations are used to measure innovation networks.

  1. In the model, it is not clear why the quadratic term and interaction terms of the core explanatory variables are included. The author should provide a rationale for their inclusion.

Reply: Thanks very much. Using the full sample data of the Yangtze River Delta urban innovation networks from 2010 to 2019 to draw a scatter diagram of the intra-city and inter-city innovation networks, we found that there is a high correlation between the two, which also confirms the above conclusion on their spatial structure similarity. At the same time, using Stata software to conduct correlation analysis, we found that the correla-tion coefficient between the two is as high as 0.8437. In order to avoid the model being affected by the correlation between the independent variables, the regression analysis of the influence of intra-city and inter-city innovation networks on regional TFP was carried out respectively.

Previous studies have shown that the quadratic term and interaction terms of the core explanatory variables will also affect innovation networks on regional growth. So, the purpose of adopting this method of the quadratic term and interaction terms of the core explanatory variables is to test the reliability of the results, and analyze the influencing mechanism in depth.

Comments on the Quality of English Language

The English writing needs further revision and refinement to meet academic standards. For example, Page 2 Line 72-73: “Third, what is the impact of intra-city and inter-city innovation networks on economic growth in the Yangtze River Delta and what is the impact mechanism?”

Reply: Thanks very much. We have revised the English writing. Page 2 Line 72-73 have been changed, “Third, what are the impact of intra-city and inter-city innovation networks on eco-nomic growth in the Yangtze River Delta and what are the impact mechanism?”

Reviewer 2 Report

This paper analyses spatial and temporal evolution characteristics of intra-city and inter-city innovation networks in 41 cities in the Yangtze River Delta in order to identify their influencing mechanisms on economic growth. The paper is well structured and the reader can follow the rationale of the essay, also because of a very plain and easy English style and grammar. However, the manuscript can be further improved by considering the following suggestions.  

Line 75: remainder, please double check this word.

Line 91: which are the three perspective?

Figure 2: it is difficult to read the maps and legends. Higher resolution is needed.

Figure 6: it would be helpful to read a description/comment of the diagram.

Line 554: after the semicolon, capital letter is not required.

English style and grammar are fine. There is a spelling mistake in line 75.

Author Response

Dear,

Thank you very much for the comments of the reviewers. We have carefully revised them. The following is our response:

Reviewer 2 Comments:

This paper analyses spatial and temporal evolution characteristics of intra-city and inter-city innovation networks in 41 cities in the Yangtze River Delta in order to identify their influencing mechanisms on economic growth. The paper is well structured and the reader can follow the rationale of the essay, also because of a very plain and easy English style and grammar. However, the manuscript can be further improved by considering the following suggestions.  

Line 75: remainder, please double check this word.

Line 91: which are the three perspective?

Figure 2: it is difficult to read the maps and legends. Higher resolution is needed.

Figure 6: it would be helpful to read a description/comment of the diagram.

Line 554: after the semicolon, capital letter is not required.

Reply: Thanks very much. We have revised them.

Line 75: “The remainders of this paper are organised as follows: the next section presents the literature review. Section 3 describes the data and empirical variables used in an-alysing the innovation networks and economic growth. With the help of ArcGIS 10.6 and MaxDEA, section 4 analyses the spatio-temporal evolution of innovation networks and economic growth in the Yangtze River Delta. Based on regression model, section 5 discusses the mechanism of innovation networks impact on economic growth. The last section concludes and discusses the paper.”

Line 91: Two perspective.

Figure 2 and Figure 6, We have redrawn the figures and tables.

Line 554: we have revised them.

Comments on the Quality of English Language

English style and grammar are fine. There is a spelling mistake in line 75.

Reply: Thanks very much. We have revised the English writing.

Round 2

Reviewer 1 Report

I appreciate the author’s revisions. However, upon reviewing the revised version, it is observed that many of the modifications made by the author have failed to adequately address the concerns raised during the review process.

Firstly, further clarification is required regarding the construction of the network. How was the network precisely constructed, and what is the author's understanding of an innovative network? Moreover, a comprehensive explanation of the nodes and edges within the intra-city and inter-city innovation networks is necessary. In Figure 1, it was mentioned that you have finally obtained 161,766 linkages of intra-city and 86,487 linkages of inter-city innovation networks. However, it should be clarified that these numbers represent the weights of the connections rather than the actual edges of the network. I am unsure about the specific definition of your network in this context. Furthermore, it is important to note that in this round of revision, the author highlighted section 3.2 as revised. Still, upon review, I did not identify any modifications made in that section. 

Secondly, despite the extensive discussion of the sixth issue in the response letter, the initial concerns remain unresolved. It is essential to establish the necessity of constructing an innovation network. The research design in this aspect is not clear. Since the author has constructed innovation networks, a deeper understanding of their significance and implications should be provided.

The English writing needs further revision and refinement to meet academic standards.

Author Response

Dear,

Thank you very much for the comments of the reviewers. We have carefully revised them. The following is our response:

Reviewer 1 Comments:

I appreciate the author’s revisions. However, upon reviewing the revised version, it is observed that many of the modifications made by the author have failed to adequately address the concerns raised during the review process.

Firstly, further clarification is required regarding the construction of the network. How was the network precisely constructed, and what is the author's understanding of an innovative network? Moreover, a comprehensive explanation of the nodes and edges within the intra-city and inter-city innovation networks is necessary. In Figure 1, it was mentioned that you have finally obtained 161,766 linkages of intra-city and 86,487 linkages of inter-city innovation networks. However, it should be clarified that these numbers represent the weights of the connections rather than the actual edges of the network. I am unsure about the specific definition of your network in this context. Furthermore, it is important to note that in this round of revision, the author highlighted section 3.2 as revised. Still, upon review, I did not identify any modifications made in that section.

Reply: Thanks very much. Innovation network refers to the cooperation of innovation actors such as government, enterprises, universities, research institutions and intermediary service agencies in technology research and development. The innovation network we understand refers to the sum of innovation linkages among different in-novation actors, in this study, the joint patent application data for the target region were selected to depict the intra-city and inter-city innovation networks, which were sourced from the incoPat database (https://www.incopat.com/). When two or more actors cooperate to apply for a patent, we think that there is an innovation linkage between them, and then a city-wide linkages sum is formed, which we call the city innovation network.

We have finally obtained 161,766 linkages of intra-city and 86,487 linkages of inter-city innovation networks, and these numbers represent the weights of the connections. What is emphasized in this study is its connection number, and it can be used to characterize the innovation network, which has been confirmed in previous studies.

We think that intra-city innovation network refers to the sum of innovation linkages amone different innovation actors in a city, while inter-city refers to the sum of innovation linkages among innovation actors in different cities, the former is within the city, and the latter is the across the city boundary.

section 3.2 was revised, “Patent literature is the largest source of technical information in the world. The report of the World Intellectual Property Organization (WIPO) shows that about 90-95 percent of global R&D outputs is contained in patents, and the rest is embodied in scientific literature, such as papers and publications[28]. Patent literature offers the ad-vantages of openness, timeliness, detailed content, and easy comparison between in-dustrial technologies or different spaces, and has become an important data source for studying knowledge production and innovation activities[29]. A joint patent application is an interactive innovation process based on knowledge sharing between organisa-tions and resource integration embedded in social networks. Economic geographers have widely recognised that joint patent application data can be researched on inno-vation networks and knowledge spillovers[30], the invention patent represents the original technology, which can better reflect the technological innovation achievements. The innovation network refers to the sum of innovation linkages among different in-novation actors, in this study, the joint patent application data for the target region were selected to depict the intra-city and inter-city innovation networks, which were sourced from the incoPat database (https://www.incopat.com/).

The main steps in constructing innovation networks were as follows: (i) Time se-lection. Because it takes 18 months for patent applications to be published, our research limited the patent application dates to the period from January 1, 2010 to December 31, 2019, and the locations to Shanghai, Anhui, Zhejiang and Jiangsu Province. The patents with 1 number of applicants were filtered, and individuals were filtered in the appli-cant category to obtain 256,934 joint patent applications in the Yangtze River Delta; (ii) Matching the innovation actors and geographical location. We matched the geo-graphical position of the applying entities by using the enterprise database of Tian-yanchao (https://www.tianyancha.com/search), and cross-processed the data of three or more actors, thus obtaining 400,201 linkages of city innovation networks; (iii) The classification of innovation networks. We divided the innovation networks within the Yangtze River Delta into intra-city and inter-city ones, and finally obtained 161,766 linkages of intra-city and 86,487 linkages of inter-city innovation networks (Figure 1). Intra-city innovation network refers to the sum of innovation linkages amone different innovation actors in a city, while inter-city refers to the sum of innovation linkages among innovation actors in different cities, the former is within the city, and the latter is the across the city boundary.”

Secondly, despite the extensive discussion of the sixth issue in the response letter, the initial concerns remain unresolved. It is essential to establish the necessity of constructing an innovation network. The research design in this aspect is not clear. Since the author has constructed innovation networks, a deeper understanding of their significance and implications should be provided.

Reply: Thanks very much. The innovation network we understand refers to the sum of innovation linkages among different in-novation actors, in this study, the joint patent application data for the target region were selected to depict the intra-city and inter-city innovation networks. These linkages represent the innovation network, which has been confirmed in previous studies. We also know that innovation network includes not only patent cooperation, but also paper cooperation, etc. However, from the existing research, patent data has been proved to be able to represent innovation network to the maximum extent. Therefore, this paper chooses the patent joint application to measure innovation network.
